# Volatiles from French and Croatian Sea Fennel Ecotypes: Chemical Profiles and the Antioxidant, Antimicrobial and Antiageing Activity of Essential Oils and Hydrolates

**DOI:** 10.3390/foods13050695

**Published:** 2024-02-24

**Authors:** Olivera Politeo, Pavao Ćurlin, Petra Brzović, Killian Auzende, Christian Magné, Ivana Generalić Mekinić

**Affiliations:** 1Department of Biochemistry, Faculty of Chemistry and Technology, University of Split, R. Boškovića 35, HR-21000 Split, Croatia; olivera@ktf-split.hr (O.P.); pavao.curlin@ktf-split.hr (P.Ć.); 2Department of Food Technology and Biotechnology, Faculty of Chemistry and Technology, University of Split, R. Boškovića 35, HR-21000 Split, Croatia; petra.brzovic@ktf-split.hr; 3Laboratoire Géoarchitecture_Territoires, Biodiversité, Urbanisme, Environnement, Université de Bretagne Occidentale, 6 Avenue Victor Le Gorgeu, CS 93837, CEDEX 3, 29238 Brest, France; killian.aurende@univ-brest.fr

**Keywords:** *Crithmum maritimum*, essential oil, hydrolate, volatile organic compounds, GC-MS, antioxidants, antimicrobials, antiageing agents

## Abstract

Sea fennel is a halophytic plant rich in valuable nutritional components and is characterized by pleasant organoleptic properties. While its essential oils (EOs) are well investigated, there are no reports on the volatiles from their corresponding hydrolates, which are the main by-products of EO isolation, as well as on their biological activity. Therefore, the composition and biological activities of EOs and corresponding hydrolates of sea fennel from Atlantic (French, FRA) and Mediterranean (Croatian, CRO) ecotypes were investigated and compared. The EO from the CRO sample was characterized by an abundance of sabinene and limonene, while that from the FRA ecotype was rich in dillapiol and carvacryl methyl ether. The CRO hydrolate was rich in terpinen-4-ol and 10-(acetylmethyl)-3-carene, while dillapiol, thymyl methyl ether and *γ*-terpinene were the main compounds in the FRA sea fennel hydrolate. The biological activities of the EOs and hydrolates were evaluated for their antioxidant (with DPPH, NO, FRAP and ORAC bioassays), antimicrobial (against some Gram+ and Gram- spoilage bacteria) and antiageing (tyrosinase, elastase and collagenase inhibition) activities. Both EOs showed low reducing powers and antiradical activities while the ability of both hydrolates to quench NO was slightly higher (35–39% if inhibition). The FRA EO showed low activity against *Staphylococcus aureus* (8 mm), while CRO moderately inhibited the growth of *P. aeruginosa* (8 mm), but strongly inhibited the other two bacterial strains. While the French EO showed no antityrosinase and anticollagenase activity, the Croatian oil significantly inhibited both enzymes (IC_50_ of 650 µg/mL and IC_50_ of 2570 µg/mL, respectively) probably due to the dominance of limonene and sabinene. Neither EO exhibited antielastase properties, while the hydrolates from both ecotypes showed no antiageing activity, regardless of the enzyme tested. The EOs from the aerial parts of sea fennel from FRA and CRO differed greatly in composition, resulting in different activities. The Croatian samples appeared to have better biological properties and are therefore good candidates for applications as preservatives or antiageing agents.

## 1. Introduction

Essential oils (EOs) are complex mixtures of volatiles extracted from aromatic plants by distillation, in most cases by hydrodistillation or steam distillation [1,2,3]. Hydrodistillation is the traditional and most commonly used method for the isolation of EOs from aromatic plants, as it is simple and inexpensive, but also environmentally friendly and non-toxic [4]. In hydrodistillation, the plant material is completely immersed in water until it boils, and the EO and water vapour are simultaneously condensed and collected [5].

During EO isolation, small amounts of the EO components, volatile hydrophilic compounds that can form weak hydrogen bonds, escape into the distillation water stream and are further condensed. The resulting aqueous solutions are called hydrolate, hydrosol, hydroflorate, aromatic water, flower water, essential water, floral water or distillate water. [1,2,3,5,6] These are complex, highly dilute, acidic aqueous mixtures containing mainly oxygenated and hydrophilic volatile components that contribute to the aroma [1,6]. The distribution of a compound between the EO and hydrolate largely depends on its solubility [6], but the losses of oxygenated hydrophilic components of the EO that migrate into the hydrolate make the flavour of the primary EO incomplete [3], but is also responsible for the fact that the composition of hydrolates often differs from the corresponding EOs, with hydrophobic isoprenoid compounds such as hydrocarbons being absent in most cases [1]. In cases where these compounds dominate in EOs, the compositions of the hydrolates are, therefore, very different from the corresponding EOs [3]. The benefits of hydrolates are related to the presence of polar or partially miscible water volatiles that migrate from the EOs, and due to the chemical characteristics of these compounds, they are recognised for their good biological activity such as antioxidant, antimicrobial, etc. [6].

Due to the increasing consumer demand for natural and safe products, EOs have recently been widely investigated as preservatives, flavour and aromatic compounds, pharmaceutic and therapeutic agents, etc. [7]. Hydrolates are also becoming increasingly popular due to their pleasant organoleptic properties which is why they are widely used across different industries. Recent studies have reviewed the characteristics and potential applications of hydrolates in cosmetology and perfumery, aromatherapy, folk medicine and the food industry [1,6,7,8,9,10]. In addition, hydrolates as by-products of EO isolation, which are usually treated as waste, are less harmful to human health compared to EOs and can, therefore, be used as natural and cheap raw materials with a wide range of applications in various industries [2].

The low number of research papers on hydrolates compared to EOs is presented in Figure 1, which shows the number of publications per year (until August 2023) on EOs and hydrolates from the Scopus database. The search criteria were “article title, abstract, keyword; essential oil and hydrosol/hydrolate“. However, it can be noted that the number of publications on hydrolates has increased over the last decade and continues to increase (Figure 1), although it is relatively small compared to studies on EOs.

Sea fennel is a wild perennial halophytic plant from the Apiaceae botanical family, characterised by the presence of valuable nutritional components such as vitamin C, minerals, phenolics, fatty acids, etc., as well as a high content of EOs with a pleasant flavour characterised by aromatic notes of fennel, celery and citrus peel [11,12,13,14,15].

Previous studies on sea fennel’s EOs and its volatiles confirmed a high geographical variability in sea fennel EOs. The main difference has been found in the presence and content of dillapiole, so that, according to Pateira et al. [16], there are two different chemotypes of the plant. Moreover, according to Renna [17], there are four main chemotypes of sea fennel based on the compounds prevalent in the EOs, namely aromatic monoterpenes, monoterpene hydrocarbons, phenylpropanoids and their intermediate forms. While the EOs from sea fennel in Mediterranean populations are well studied, like those from Turkey [18,19,20,21,22], Italy [23,24,25], Tunisia [12], Spain [26], Algeria [27], Greece [28] and France [29], there are few reports on volatiles from sea fennel populations on the Atlantic coasts of Portugal [9,13,16,26,30] and France [24,29,31].

EOs from Mediterranean sea fennel are well documented, whereas there are few studies on the Atlantic sea fennel ecotype, and there are no reports on the volatiles within their corresponding hydrolates or on their biological activity. Therefore, the present study focuses on: (a) investigating and confirming the differences in the EO composition of typical Atlantic (French) and Mediterranean (Croatian) sea fennel ecotypes; (b) comparing the volatiles in the corresponding hydrolates and identifying correlations or links to the EOs; and (c) investigating and comparing the biological properties of the samples, both EOs and hydrolates, in terms of antioxidant, antimicrobial and antiageing activity.

## 2. Materials and Methods

### 2.1. Plant Material

About 2 kg of the aerial parts of the sea fennel were collected in October 2022 from two sampling sites; the Mediterranean ecotype was harvested in Dalmatia (43°39′44″ N 15°56′40″ E, Croatia), while the Atlantic ecotype was collected on the Brittany shoreline near Brest (48°21′47″ N 4°31′51″ O, France). The plant material was identified by the Tonka Ninčević Runjić (Institute for Adriatic Crops and Karst Reclamation, Split, Croatia) (CRO sample) and by Christian Magné (Université de Bretagne Occidentale, Brest, France) (FRA sample). The voucher specimens were also deposited in the herbariums of the authors’ two institutions.

The samples were air-dried for 15–20 days in an aerated and shaded place before EO extraction.

### 2.2. Hydrodistillation

The EOs were isolated from dry plant materials by hydrodistillation (Clevenger apparatus, 3 h) following the procedure reported by Bektašević et al. [32]. The separated EOs were dried over anhydrous Na_2_SO_4_. Both the EOs and the hydrolates were stored at −20 °C until analysis [33].

### 2.3. Chemical Analysis

The volatiles from the hydrolates were isolated by headspace solid-phase microextraction (HS-SPME) using DVB/CAR/PDMS-coated fibres (50/30 µm, Supelco, Sigma Aldrich, Bellefonte, PA, USA) according to the procedure described by Politeo et al. [33]. After thermostation (at 40 °C for 30 min), the fibre was introduced into the injection port of the gas chromatograph for thermal desorption (3 min).

EOs and hydrolates were analysed by gas chromatography–mass spectrometry (GC-MS) using a gas chromatograph (model 8890, Agilent Inc., Santa Clara, CA, USA) equipped with an automatic liquid injector (model 7693A, Agilent Inc.) and a mass spectrometer (model 7000D GC/TQ, Agilent Inc.) using the HP -5MS UI column (30 m × 0.25 mm, 0.25 µm, Agilent Inc.). The method conditions applied for the analysis of EOs and hydrolates were described in our previous study [33], and the detected compounds were identified by comparing their retention indices with a series of *n*-hydrocarbons (C8–C40, Supelco Inc., Sigma Aldrich) and their mass spectra with reference data from the Wiley 7 MS library (Wiley, NY, USA) and the NIST02 (Gaithersburg, MD, USA) database, as well as with literature reports [34]. The results are reported as mean ± standard deviation of two injections.

### 2.4. Biological Activity

#### 2.4.1. Antioxidant Activity

The antioxidant activity was tested spectrometrically using a microplate reader (Synergy HTX Multi-Mode Reader, BioTek Instruments, Inc., Winooski, VT, USA) and a SPECORD 200 Plus, Edition 2010 (Analytik Jena AG, Jena, Germany).

The reducing activity of the samples was tested using the Ferric Reducing Antioxidant Power (FRAP) assay [35] and the results are expressed in µM Fe^2+^/L.

Three different radicals were used in testing the radical scavenging activity of the samples: the synthetic stable 2,2-diphenyl-1-picrylhydrazyl radical (DPPH•) and two biologically important radicals: the nitric oxide radical (NO•) and the hydroperoxyl radical (HOO•).

DPPH inhibition activity was measured according to Katalinić et al. [36], while NO- radical inhibition was measured as described by Dastmalchi et al. [37]. The radical inhibition results are expressed as a percentage (%).

Oxygen Radical Absorbance Capacity (ORAC) assay was performed according to the method reported by Čagalj et al. [38], and the final results of inhibition of oxidation induced by peroxyl radicals are expressed in µM Trolox equivalents (µM TE).

For the FRAP, DPPH and NO assays, pure hydrolates and essential oils diluted up to a concentration of 1 mg/mL were tested, while for the ORAC measurements, EOs were tested at 10-times dilution and hydrolates at 100-times dilution. All measurements were performed with five replicates.

#### 2.4.2. Antimicrobial Activity

The antibacterial activity of EOs and hydrolates from sea fennel against one Gram-negative (*Staphylococcus aureus* ATCC 33862) and two Gram-positive (*Escherichia coli* ATCC 1053 and *Pseudomonas aeruginosa* ATCC 27853) bacterial strains was investigated using the disk diffusion method [39].

Each bacterium was grown at 37 °C for 18 h on Tryptone Soy Broth (TSB) medium and turbidity was adjusted to 10^8^ CFU/mL. Seventy microliters of the bacterial suspension was then spread onto a 9 cm Petri dish containing 15 mL of sterile Mueller Hinton Broth medium with 20 g/L agar. Paper disks (Whatman 3MM, Whatman, Inc., Clifton, NJ, USA) with a diameter of 5.5 mm were placed in the dish and soaked with 7.5 µL of EOs or hydrolates (reduced volume due to the small sample amount) at different concentrations. Three replicates of each concentration were performed. A positive control dish contained only TSB medium inoculated with a bacterial suspension (10^8^ CFU/mL), while 7.5 µL of antibiotics (5 mg/mL streptomycin) were added to the suspension of microorganisms in the negative control. The dishes were sealed with parafilm and, after incubation (24 h at 30 °C), the inhibition diameter was measured.

#### 2.4.3. Antiageing Activity

Antityrosinase activity was evaluated using L-tyrosine as substrate and mushroom tyrosinase (SIGMA) according to the slightly modified method described by Masuda et al. [40]. The samples (plant EOs or hydrolates) were dissolved in DMSO (50%, *v*/*v*). Then, 40 μL of each sample was mixed with 80 μL phosphate buffer (0.1 M, pH 6.8), 40 μL tyrosinase (30 units/mL in phosphate buffer, pH 6.5) and 40 μL 2.5 mM L-tyrosine on a 96-well microplate. The absorbance of the sample was monitored kinetically at 475 nm for 15 min, taking a reading every 30 s, and compared to a blank containing all ingredients except EOs or hydrolates. The inhibition (%) of catalysis of L-tyrosine to L-dopa and then to dopaquinone was calculated using the following formula:$$Inhibition\left(\%\right)=[1-\left(\frac{At-A0}{Ct-C0}\right)]\times 100$$
where At and A0 are the absorbances of the sample at t and t = 0, respectively, and Ct and C0 are the absorbances of the control at t and t = 0, respectively. Finally, the IC_50_ (concentration of EO or hydrolate causing 50% enzyme inhibition) was determined from the regression curve of inhibition percentages.

Antielastase was assessed using elastase from porcine pancreas Type IV E0258 (Sigma) and N-Succinyl-Ala-Ala-Ala-*p*-nitroanilide (SANA) as substrate according to the method by Kalyana Sundaram et al. [41]. Twenty-five µL of the EOs or hydrolates at different concentrations in DMSO (5%, *v*/*v*) (3 replicates) were mixed with 175 µL of Tris-HCl buffer (0.2 M, pH 8) and 25 µL of enzyme (10 µg/mL) in each well of a microplate. After incubation (30 min at 37 °C), the reaction was initiated by adding 25 µL of SANA (1 mg/mL). Catalysis of SANA in *p*-nitroaniline was monitored for 15 min at 410 nm with a reading every 30 s. The negative control consisted of DMSO (5%, *v*/*v*) instead of the EOs, and the percentage of elastase inhibition (%) was calculated according to the following formula:$$Inhibition(\%)=(1-\frac{Ss}{Sc})\times 100$$
where Ss and Sc are the slopes in the sample and control assays, respectively. Finally, the IC_50_ was determined using the regression curve of the inhibition percentage.

The anticollagenase activity of the EOs and hydrolates was investigated using the Enzo Life Sciences MMP-1 kit (Colorimetric Drug Discovery Kit BML-4K404, Enzo Life Sciences, Farmingdale, NY, USA). Each microplate well contained 50 µL of buffer (50 mM HEPES, 10 mM CaCl_2_, 0.05% Brij-35, 1 mM DTNB, pH 7.5), 20 µL of the samples diluted in the buffer at various concentrations (3 replicates) and 20 µL of the enzyme at 637.5 mU/µL. The reaction was initiated by adding the chromogenic substrate Thiopeptide (Ac-PLG-[2-mercapto-4-methyl-pentanoyl]-LG-OC_2_H_5_) (10 µL at a concentration of 1 mM). The catalysis of thiopeptide to 2-nitro-5-thiobenzoic acid was monitored kinetically at 412 nm for 15 min, with a measurement every 30 s. The reaction rate was determined using the linear part of the kinetics (from 0 to 10 min). Controls were run with kit buffer without EO or hydrolate, and collagenase inhibition (%) was calculated using the following formula:$$Inhibition\left(\%\right)=\left(1-\frac{Ss}{Sc}\right)\times 100$$
where Ss and Sc are the slopes of the sample and control assays, respectively. Finally, the IC_50_ was determined using the regression curve of inhibition percentage.

### 2.5. Statistical Analysis

The data obtained (expressed as mean ± standard deviation) were statistically analysed using R 4.1.0 software (www.cran.r-project.org, accessed on 13 January 2024). Comparisons between data were performed using the Tukey post hoc test with statistical significance set at *p* < 0.05.

## 3. Results and Discussion

### 3.1. Chemical Composition of EOs and Hydrolates

#### 3.1.1. Essential Oils

The composition of the sea fennel oils studied is shown in Table 1. A total of 13 compounds were detected in the French sea fennel, while 12 compounds were found in the Croatian sample. The main compounds in the French sea fennel EO were dillapiole (62.10 ± 1.83%), carvacryl methyl ether (18.00 ± 2.40%) and *γ*-terpinene (9.88 ± 1.10%) (89.98% of the total identified components), while the Croatian sea fennel EO was rich in sabinene (51.47 ± 3.22%) and limonene (36.28 ± 2.99%) (87.75% of the total identified components). It is clear from the results that the sea fennel samples studied are from completely different chemotypes, with the main difference being the presence or absence of dillapiole. Based on the results of previous studies, dillapiole was found to be a component of sea fennel EO in samples from Turkey [19,20,21], Italy [24], Tunisia [12], Greece [28], France [24,29,31] and Portugal [9,13,16], while it was never detected in Croatian samples [33,42,43,44,45].

However, few studies have reported that it is one of the dominant EO components. Senatore et al. [19] reported different EO compositions and chemical profiles of wild sea fennel at two sites in southern Turkey. While the presence of dillapiole could not be confirmed in the sample from Antalya, its content was significantly high (20.6%) in the sample from Mersin. Similar results were obtained by Özcan et al. [20], who also found variations in the dillapiole content in crops harvested in different years from the same site (1.9% in the first year and 21% in the second year). Jallali et al. [12] also found high dillapiole contents in samples from Tunisia (14.3 and 40.3%) by.

As is known by the authors, the presence of dillapiole has been confirmed in all sea fennel samples from the Atlantic coast, either from Portugal [13,14] or from France [24,29,31]. Pateira et al. [16] reported variations in the composition of sea fennel EOs from plant samples at three stages of development (vegetative, flowering and fruiting), from different locations and from three years, with dillapiole content ranging from 0.3 to 46.4%. On the other hand, the samples from the French Atlantic coast had the highest dillapiole concentration, with 17.5% [31], 25% [29] and 55.7% [24].

As mentioned above, the other two dominant compounds in the studied sample from the Atlantic coast were carvacryl methyl ether (18%) and *γ*-terpinene (≈10%). The results from Pavela et al. [24] for Atlantic sea fennel from France showed a thymyl methyl ether content of 11.8% and a *γ*-terpinene content of 14.0%, and Pavela et al. [31] showed a thymol methyl ether content of 2.0% and a *γ*-terpinene content of 33.0%.

The results obtained for Croatian sea fennel are consistent with previously published reports, in which more or less the same EO compounds are reported in different proportions. EOs from Croatian sea fennel were studied by Kulišić Bilušić et al. [42], Generalić Mekinić et al. [43], Politeo et al. [33] and Politeo et al. [44].

It is hard to draw general conclusions about the causes of the differences in the chemical composition of sea fennel EOs, as they may be due to the different abiotic and biotic factors that affect the synthesis and accumulation of plant secondary metabolites, such as the harvest location or geographical variations [24,29,31], plant vegetation/harvest period/growth cycle [16,20,28,39,45], plant part used [16,43,44], climatological factors [16,20], soil abiotic factors [26,39], etc. However, our results agreed with the conclusions of other authors [12,16,22], according to which the main reason is simply the existence of different plant chemotypes (intraspecific variability).

Kulišić Bilušić et al. [42] reported that limonene (58.4%), sabinene (26.5%), *γ*-terpinene (2.8%) and terpinene-4-ol (5.6%) are the main constituents of sea fennel EOs. In the research by Generalić Mekinić et al. [43] and Politeo et al. [44], the predominant compounds were the same, with slight differences in the amounts detected depending on the part of the plant analysed (leaves, flowers, stems). However, Politeo et al. [33] investigated the influence of the EO isolation method on the chemical profile of the EO and also found differences between the samples. The main EO compounds, obtained by hydrodistillation, were again limonene (51.4%) and sabinene (25.2%). These previous studies all reported a higher yield of limonene in the samples, while the sample from the present study had a higher content of sabinene. One of the reasons for this could be the when the plants were harvested, as the samples from the previous studies were collected in summer (from July to September), while the sample from this study was collected in October.

#### 3.1.2. Hydrolates

Hydrolates are complex mixtures containing traces of EO constituents and other water-soluble compounds that give them their specific sensory properties and flavour, as well as their biological activity [3]. Hydrolates generally contain polar compounds such as alcohols (monoterpenes and sesquiterpenes), aldehydes, ketones and esters, while lipophilic hydrocarbon monoterpenes with low water solubility are usually not present [3,6]. According to Aćimović et al. [3], the ratio between hydrocarbons and oxygenated compounds in the EO affect the chemical profile of their corresponding hydrolates. In general, the hydrolate has a similar composition when the parent EO is rich in oxygenated compounds, while the composition of the hydrolate differs significantly from that of the EO when hydrocarbons are the main constituents of the EO. However, the authors also reported some studies with completely different profiles of EOs and their corresponding hydrolates.

The compounds within the volatiles from the sea fennel hydrolates are shown in Table 2 where 39 different compounds can be seen, differing in occurrence and quantity. While 32 components were detected in the French sample, only 17 volatile compounds were found in the Croatian sea fennel hydrolate. Again, the greatest differences were found in the compounds that were detected in the highest proportions. The predominant components in the French sea fennel hydrolate were dillapiole (36.66 ± 5.66%), thymyl methyl ether (26.30 ± 1.85%) and *γ*-terpinene (9.34 ± 0.24%), while in the Croatian sample they were terpinen-4-ol (41.93 ± 2.99%) and 10-(acetylmethyl)-3-carene (13.80 ± 2.01%) (Table 2). The results are consistent with our previous studies on sea fennel hydrolates, where terpinen-4-ol was the predominant compound with a yield of 13.86%, while 10-(acetylmethyl)-3-carene (13.45%), (*E*)-*α*-ionone (10.04%) and (*Z*)-*β*-damascenone (4.80%) were found in large amounts among the other compounds [33].

Inouye et al. [46] reported that plant hydrolates can be categorised into several groups based on the functional group of the main constituent: alcohol, aldehyde, ketone, ester, phenol or phenyl methyl ether groups. Accordingly, the hydrolate from the French sea fennel belongs to the phenyl methyl ether group, while the hydrolate from the Croatian sea fennel belongs to the alcohol group.

The results in Table 1 and Table 2 show differences in the chemical composition of the EOs and hydrolates of the Croatian and French samples. The French sea fennel EOs and the corresponding hydrolates were rich in dillapiole, whereas the dominant components of the Croatian EOs (limonene and sabinene) were not found in its hydrolate. This is consistent with the fact that the distribution of a component between the EO and hydrolate depends primarily on its solubility in the EO itself. The French hydrolate showed a high content of dillapiole and *γ*-terpinene, but also a high content of thymyl methyl ether (26.3%), which was present in low concentrations in the corresponding EO. The reverse relationship was found for carvacryl methyl ether, which was present in the EO (18.0%) and found in a much lower amount in the hydrolate (0.28%). As expected, sabinene (bicyclic unsaturated monoterpene) and limonene (aliphatic hydrocarbon, cyclic monoterpene), compounds that dominate in Croatian sea fennel EO due to their lipophilicity, were not found in the hydrolate.

### 3.2. Biological Activities of EOs and Hydrolates

While the biological activity of EOs has been extensively studied, especially with regard to their antimicrobial or antioxidant properties, hydrolates have only recently gained attention [3,4,6]. Here, we report the results of a comparative study on the antioxidant, antimicrobial and antiageing activities of EOs and hydrolates from Atlantic and Mediterranean ecotypes of sea fennel.

#### 3.2.1. Antioxidant Activity

In this study, the antioxidant activity of the samples was tested using four different methods: FRAP, DPPH, NO and ORAC. While the FRAP method was used to evaluate the reducing activity of the samples, the other three methods provided information on the radical scavenging activity of the samples. The results obtained are shown in Table 3.

The reducing activity of all samples is almost negligible compared to the activity of gallic acid which is recognised as one of the most potent natural phenolic antioxidants. The FRAP values of the tested samples ranged between 0.77 and 1.34 µM Fe^2+^/L. Similar observations can be made for the DPPH inhibition results with an inhibition percentage of 2% for both EOs, and a slightly higher activity for the corresponding hydrolates. Gallic acid, which was tested at the same concentration, provided 95% of radical inhibition.

The results show that the antiradical activity against NO• differs between the samples. The activity of gallic acid was not significant in this method as it inhibited a similar share of radicals to the EO samples. Both EOs and the standard compound (at concentration of 1 mg/mL) showed more than three-times-lower activity than the corresponding hydrolates, with the CRO EO giving slightly better results. On the other hand, the hydrolates from FRA showed better activity than the sample from CRO.

The radical scavenging activity against peroxyl radicals was tested with the ORAC assay, which measures the degree of inhibition of oxidation triggered by peroxyl radicals. While the ORAC values of CRO and FRA sea fennel EOs did not differ significantly, the activity of the hydrolate CRO was 3.8 times higher (138 vs. 36 µM Trolox equivalents/L) than that of FRA. The results of previous studies are in agreement with these, showing that the hydrolates have a low or moderate antioxidant capacity [47], but their chemical composition is crucial for understanding the mechanisms of their biological activity [1,2].

The weak free radical scavenging activity of sea fennel EOs due to the high content of terpenes, which are not recognised as potent antioxidants, was also reported by Kulišić-Bilušić et al. [42] and Jallali et al. [12] Sharopov et al. [48] investigated the antioxidant activity of eighteen common essential oil constituents using the FRAP and DPPH methods and reported the high antioxidant activity of carvacrol, eugenol and thymol, while the activity of limonene was low in both methods. The chemical composition of CRO EO was characterised by the presence of non-oxygenated compounds (93.5%), while only a small amount of oxygenated compounds was detected, confirming previous reports on the low antioxidant activity of terpenes. In contrast, the FRA EO was richer in oxygenated compounds (80.6%), with dillapiol (62.1%) dominating, suggesting that its antioxidant activity is probably very low (no data were found in the literature), especially since it does not contain hydroxyl groups in its structure that could donate hydrogen to stabilise free radicals. In addition, the potentially good antioxidant activity of other compounds that were present in low amounts in the samples, such as α- and γ-terpinene and terpinolene [49], should not be neglected. On the other hand, all compounds detected in the CRO hydrolate were oxygenated compounds (oxygenated hydrocarbons), while represented 68% of the compounds in the FRA hydrolate, which probably affected the antiradical activity of the samples.

#### 3.2.2. Antimicrobial Activity

As can be seen (Table 4), the French EO showed no antimicrobial activity against *Escherichia coli* and *Pseudomonas aeruginosa*, but a slight activity against *Staphylococcus aureus* (8 mm). Conversely, the Croatian EO moderately inhibited the growth of *P. aeruginosa* (8 mm), but strongly inhibited that of the other two bacterial strains (18 and 25 mm for *E. coli* and *S. aureus*, respectively). Interestingly, the CRO EO’s inhibition of the latter bacterial strain was as strong as that of the commonly used bactericidal streptomycin. Furthermore, neither the French nor the Croatian hydrolates showed any antimicrobial effect.

These results can be discussed taking into account the composition of the EOs and hydrolates. As mentioned above, the CRO EO showed stronger antibacterial activity against all tested bacteria. This could be due to the predominance of monoterpene hydrocarbons in this EO [50]. In particular, the broad-spectrum bactericidal activity of limonene is well documented [51,52,53,54], which allows it to be widely used in antibacterial treatment and food preservation. In particular, the mechanism of the bactericidal effect of limonene has been elucidated [54]. A similar observation can be made for sabinene, the other main component of CRO EO [55], while terpinen-4-ol probably contributes much less to the measured antimicrobial activities [56]. Pedreiro et al. [9] also investigated the antibacterial activity of sea fennel EOs from Portugal against *E. coli* and *S. aureus*, and reported good activity, of 8.2 and 11.6 mm, respectively. According to the authors, this activity was due to the high content of *γ*-terpinene and sabinene in the EO studied.

In contrast, the FRA EOs showed weak antimicrobial activity and only slightly inhibited the growth of *S. aureus*. This can also be related to the composition of this EO, which differs greatly from that of the CRO ecotype. For example, the main component, dillapiole, was only found to have repulsive effects against mosquitos [57], but no antimicrobial activity. In addition, carvacryl methyl ether has no bactericidal effect despite its structure being close to carvacrol [58]. The very limited activity against *S. aureus* could be explained by the low content of sabinene in FRA EO.

Sánchez-Hernández et al. [59] reported the antibacterial activity of sea fennel EOs, rich in *γ*-terpinene, thymol methyl ether, and dillapiole, against *S. aureus* (13 mm) but found there was no activity against *E. coli*, which is in accordance with the results of this study. Activity against *E. coli* was also not detected in the study by Jallali et al. [12], while the diameter of inhibition against *P. aeruginosa* was 10 mm. On the other hand, the EO was more effective against the tested Gram-positive species, *B. cereus* and *S. aureus* with inhibition zones of 13 and 15 mm, respectively. Again, the tested EO was rich in *γ*-terpinene (19.3%), thymol methyl ether (20.6%) and dillapiole (40.2%).

Kunicka-Styczynska et al. [60] investigated the antibacterial activity of lavender hydrolates (from fresh or dried herbs or flowers) and found no activity, which could be due to the high dilution of the samples, which according to the authors, could also be the reason for the ineffectiveness of the sea fennel samples tested in this study.

#### 3.2.3. Antiageing Activity

The results of antityrosinase activity of EOs and hydrolates from French and Croatian ecotypes are shown in Table 5. While the EO from the sea fennel from France showed no antityrosinase activity, the Croatian EO inhibited mushroom tyrosinase (IC_50_ = 649 µg/mL), although less than the arbutin standard. Since the main components of CRO EO are limonene and sabinene and these compounds have strong antityrosinase properties in other plant EOs [61,62], it is likely that the strong activity measured in CRO EO is due to the abundance of these two monoterpene hydrocarbons. Interestingly, the same observations could be made for anticollagenase activity, where only the CRO EO inhibited the enzyme (IC_50_ = 2571 µg/mL). Again, the anticollagenase activity of the two terpenoids limonene and sabinene was recently reported [63]. These values for anti-ageing activity could be considered low, but were obtained after a huge dilution (1000-fold) of the pure EOs. The EOs from the French and Croatian ecotypes of sea fennel showed no antielastase properties. Finally, the hydrolates from both ecotypes showed no antiageing activity, regardless of the enzyme used. Antimicrobial and antioxidant compounds in hydrolates are important to prevent the oxidation of ingredients in various cosmetic products and to maintain their quality, safety and shelf life. Their advantages also lie in the fact that they are of natural origin and can be used for flavouring due to the presence of compounds with pleasant aroma properties [6].

## 4. Conclusions

The investigated essential oils from the Mediterranean (chemotype II and monoterpene hydrocarbon chemotype) and Atlantic (chemotype I and phenylpropanoid chemotype) sea fennel populations differ in their chemistry and the presence/content of the major constituents, as well as the corresponding hydrolates, which mainly contain oxygenated compounds. The predominant compound in the FRA EO was dillapiole (62.10%), while sabinene (51.47%) and limonene (36.28%) were detected in the highest amounts in the Croatian sample. While dillapiole had the highest percentage in the FRA hydrolate, sabinene was the most abundant constituent of the CRO hydrolate. Although the antioxidant activity of both EOs and hydrolates was not appreciable, the antimicrobial activity of the CRO EO against *S. aureus* and *P. aeruginosa* is comparable to that of streptomycin. Probably due to the high content of limonene and sabinene, the CRO EO also showed strong antityrosinase and anticollegenase inhibitory activity, while the FRA EO showed no activity, underlining its potential for use in the cosmetics industry. Given the good chemical composition and biological activities of the Croatian sea fennel chemotype, further studies directed on its isolates and their application should be conducted. Hydrolates, as by-products of EO isolation, are usually treated as a waste and are disposed of. However, they can serve as natural, valuable and cost-effective raw materials with a wide range of applications that will also have a positive impact on the environment.

## Figures and Tables

**Figure 1 foods-13-00695-f001:**
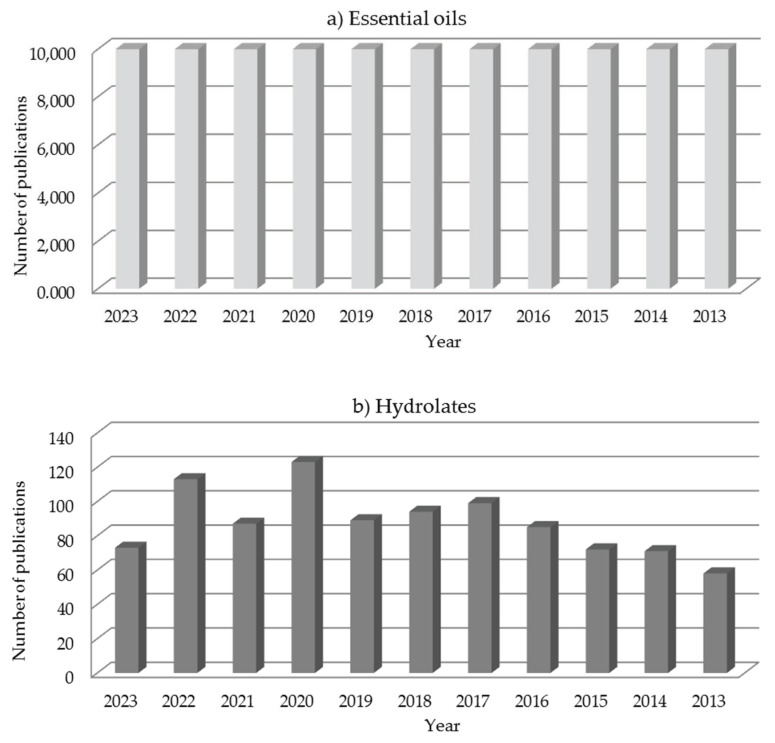
Number of publications per year on (**a**) essential oils and (**b**) hydrolates; obtained using search in Scopus database.

**Table 1 foods-13-00695-t001:** GC-MS chemical composition of French (FRA) and Croatian (CRO) sea fennel essential oils.

RI	Compounds	FRA (%)	CRO (%)
921	*α*-thujene	0.15 ± 0.02	nd
934	*α*-pinene	0.84 ± 0.13	tr
970	Sabinene	4.24 ± 0.10	51.47 ± 3.22
992	*β*-pinene	0.17 ± 0.02	0.89 ± 0.02
1012	*α*-terpinene	0.20 ± 0.02	0.98 ± 0.03
1020	*p*-cymene	2.97 ± 0.43	tr
1039	(*Z*)-*β*-ocimene	0.80 ± 0.17	nd
1032	Limonene	nd	36.28 ± 2.99
1042	(*E)*-*β*-ocimene	nd	tr
1056	*γ*-terpinene	9.88 ± 1.10	3.49 ± 0.07
1065	*cis*-sabinene hydrate	nd	0.10 ± 0.01
1086	Terpinolene	0.12 ± 0.01	0.37 ± 0.05
1118	*cis*-*p*-menth-2-en-1-ol	nd	0.10 ± 0.01
1176	terpinen-4-ol	0.25 ± 0.01	5.35 ± 0.04
1232	thymyl methyl ether	0.21 ± 0.01	nd
1242	carvacryl methyl ether	18.00 ± 2.40	nd
1620	Dillapiole	62.10 ± 1.83	nd
	TOTAL	99.93	99.03

RI = retention indices on HP-5MS UI column, tr—traces (<0.1%), nd—not detected.

**Table 2 foods-13-00695-t002:** GC-MS chemical composition of French (FRA) and Croatian (CRO) sea fennel hydrolates.

RI	Compounds	FRA (%)	CRO (%)
921	*α*-thujene	0.40 ± 0.02	nd
934	*α*-pinene	1.36 ± 0.02	nd
970	sabinene	5.64 ± 0.07	nd
986	2,3-dehydro-1,8-cineole	tr	nd
989	*β*-myrcene	0.61 ± 0.07	nd
998	octanal	tr	nd
1001	*α*-phellandrene	tr	nd
1012	*α*-terpinene	0.62 ± 0.04	nd
1020	*p*-cymene	6.40 ± 0.07	nd
1024	*β*-phellandrene	0.19 ± 0.01	nd
1039	(*Z*)-*β*-ocimene	2.10 ±0.01	nd
1041	benzeneacetaldehyde	0.77 ± 0.15	5.06 ± 0.03
1056	*γ*-terpinene	9.34 ± 0.24	nd
1065	*cis*-sabinene hydrate	tr	nd
1086	terpinolene	0.35 ± 0.02	nd
1095	*trans*-sabinene hydrate	tr	4.73 ± 0.04
1097	linalool	nd	tr
1118	*cis*-*p*-menth-2-en-1-ol	0.25 ± 0.04	4.26 ± 0.02
1138	*trans*-*p*-menth-2-en-1-ol	tr	3.37 ± 0.05
1176	terpinen-4-ol	2.12 ± 0.22	41.93 ± 2.99
1183	*p*-cymen-8-ol	tr	nd
1188	*α*-terpineol	0.21 ± 0.01	5.71 ± 0.51
1205	*trans*-pipertiol	nd	2.46 ± 0.01
1218	*trans*-carveol	0.20 ± 0.01	3.36 ± 0.03
1222	*cis*-carveol	nd	3.75 ± 0.02
1232	thymyl methyl ether	26.30 ± 1.85	nd
1242	carvacryl methyl ether	0.28 ± 0.02	nd
1293	thymol	0.32 ± 0.03	tr
1301	carvacrol	tr	1.07 ± 0.02
1312	*p*-vinylguaiacol	0.76 ± 0.04	nd
1327	myrtenyl acetate	nd	0.99 ± 0.01
1384	(*E*)-*β*-damascenone	nd	1.03 ± 0.09
1390	10-(acetylmethyl)-3-carene	nd	13.80 ± 2.01
1422	dihydrodehydro-*β*-ionone	nd	5.88 ± 0.09
1498	bicyclogermacrene	0.23 ± 0.01	nd
1521	myristicin	0.48 ± 0.00	nd
1557	elemicin	tr	nd
1563	germacrene B	0.39 ± 0.02	nd
1620	dillapiole	36.66 ± 5.66	1.52 ± 0.01
	**TOTAL**	95.98	98.92

RI = retention indices on HP-5MS UI column, tr—traces (<0.1%), nd—not detected.

**Table 3 foods-13-00695-t003:** Antioxidant activity of the French (FRA) and Croatian (CRO) sea fennel essential oils (EOs) and hydrolates.

Antioxidant Assay	FRA EO	CRO EO	FRA Hydrolate	CRO Hydrolate	Gallic Acid (Standard)
FRAP (µM Fe^2+^/L)	1.34 ± 0.17 c	1.11 ± 0.08 b	0.77 ± 0.19 a	1.16 ± 0.15 b	2485.09 ± 3.73 *
DPPH (% inhibition)	1.99 ± 0.16 a	2.00 ± 0.17 a	2.39 ± 0.25 b	2.17 ± 0.13 ab	95.35 ± 0.33 **
NO (% inhibition)	10.11 ± 0.14 a	11.30 ± 0.24 b	38.94 ± 0.13 d	35.20 ± 0.19 c	11.56 ± 0.90 **
ORAC (µM Trolox equivalents/L)	115.77 ± 0.35 bc	95.48 ± 3.82 b	36.22 ± 2.86 a	138.05 ± 0.41 c	-

* at concentration 0.1 mg/mL; ** at concentration 1 mg/mL. In each line, values followed by different letter are statistically significant (*p* < 0.05, by Tukey’s test).

**Table 4 foods-13-00695-t004:** Antimicrobial activity of the French (FRA) and Croatian (CRO) sea fennel essential oils (EOs) and hydrolates, expressed as inhibition diameter (mm).

Bacterial Strains	FRA EO	CRO EO	FRA Hydrolate	CRO Hydrolate	Streptomycin
*Escherichia coli*	Ø	18.0 ± 1.3 ^b^	Ø	Ø	28.5 ± 2.1 ^a^
*Staphylococcus aureus*	8.0 ± 0.5 ^b^	24.8 ± 3.3 ^a^	Ø	Ø	22.3 ± 1.0 ^a^
*Pseudomonas aeruginosa*	Ø	8.3 ± 0.6 ^b^	Ø	Ø	26.6 ± 1.5 ^a^

Data are means of inhibition diameter (mm) ± S.D. of three replicates. In each line, values followed by different letter are statistically significant (*p* < 0.05, by Tukey’s test); Ø no inhibition activity.

**Table 5 foods-13-00695-t005:** Antiageing activities of the French (FRA) and Croatian (CRO) sea fennel essential oils (EOs) and hydrolates, as well as of positive controls. Antityrosinase, anticollagenase, and antielastase are expressed as IC_50_ in µg/mL.

Enzymes	FRA EO	CRO EO	Arbutin	EGCG	Ursolic Acid
Tyrosinase	Ø	649 ± 54	137 ± 6	/	/
Collagenase	Ø	2571 ± 334	/	51 ± 7	/
Elastase	Ø	Ø	/	/	238 ± 13

EGCG–epigallocatechin gallate. Data are means of IC50 ± S.D. of three replicates. Ø, no inhibition activity.

## Data Availability

The original contributions presented in the study are included in the article, further inquiries can be directed to the corresponding author.

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
