# Peer review of "Volatiles from French and Croatian Sea Fennel Ecotypes: Chemical Profiles and the Antioxidant, Antimicrobial and Antiageing Activity of Essential Oils and Hydrolates"

_foods, 2024, doi:10.3390/foods13050695_

Round 1
Reviewer 1 Report
Comments and Suggestions for Authors
The manuscript by Politeo et al. focuses on the evaluation of chemical profile of volatiles from different sea fennel ecotypes. Additionally, antioxidant, antimicrobial and anti-aging effect of the extracts have been assessed. The authors reported very potent hydrates in terms of their potential as antioxidant and antimicrobial agents. The manuscript brings new perspectives to the field and makes a contribution to the existing literature. The language and structure of the manuscript are clear and easily understandable. The abstract provides a comprehensive overview of the study's objectives, methodology, and findings. Introduction provide sufficient background and include all relevant references. However, there are some very minor clarifications required.
The comments for authors:
1. Moisture content of the dried sample material has to be provided.
2. Have you done the chemical profile analysis in duplicate or triplicate? SD values should be provided in Table 1 and 2.
3. Conclusion should be more supported by the results.
4. More recent studies should be cited.
5. The text should be checked for typos.
Author Response
Reviewer #1
The manuscript by Politeo et al. focuses on the evaluation of chemical profile of volatiles from different sea fennel ecotypes. Additionally, antioxidant, antimicrobial and anti-aging effect of the extracts have been assessed. The authors reported very potent hydrates in terms of their potential as antioxidant and antimicrobial agents. The manuscript brings new perspectives to the field and makes a contribution to the existing literature. The language and structure of the manuscript are clear and easily understandable. The abstract provides a comprehensive overview of the study's objectives, methodology, and findings. Introduction provide sufficient background and include all relevant references. However, there are some very minor clarifications required.
Response: We would like to thank the Reviewer for the time, efforts and positive comments. We have considered all comments and suggestions and the manuscript has been modified accordingly.
The comments for authors:
- Moisture content of the dried sample material has to be provided.
Response: Thank you for your comment. Unfortunately, we did not determine the moisture content of the samples prior to EO isolation, and we found that this determination is not useful due to the long storage time of the plant material. We will certainly take this into account in our next study.
- Have you done the chemical profile analysis in duplicate or triplicate? SD values should be provided in Table 1 and 2.
Response: The GC-MS analysis of the samples was performed in duplicate and statistical deviations are now provided in both Tables. Thank you for the comment.
- Conclusion should be more supported by the results.
Response: Following the comment of the Reviewer, we improved the Conclusion section. Thank you.
- More recent studies should be cited.
Response: The entire manuscript has been supplemented by several new/recent studies on this topic.
- The text should be checked for typos.
Response: The whole manuscript has been checked and this has been corrected in the paper according to the Reviewer comment.
Reviewer 2 Report
Comments and Suggestions for Authors
General remarks:
The manuscript's authors aim to study the chemical diversity and biological activity of the essential oils and hydrolates obtained from aerial parts of different Crithmum maritimum ecotypes. The authors concluded that hydrolate from Croatian sea fennel is a good candidate for cosmetic applications as an anti-aging or preservative agent. The manuscript must be significantly reviewed.
Abstract
The abstract should be more precise and better organized. The aim of the study needs to be more accurate.
Introduction
The manuscript's introduction must be restructured with information relevant to the study's objectives. For example, the comparative analysis of publications studying essential oils vs. hydrolates, presented in the manuscript, needs to be sufficiently justified in the context in which the properties of hydrolases (in general) are comprehended. Hydrolates already have several applications in the cosmetic and pharmaceutical industries. Please revise.
Materials and method
How was the plant material used identified? Is there a voucher for each species studied? Where is it stored? Also, the authors should include information about the statistical analysis performed.
Results and discussion
The scientific level could be better throughout the manuscript, especially in interpreting the results. Although the authors declare MIC determination (lines 162-164), the results are not in the results and discussions section. Please review! Also, the authors need to explain the causes of the recorded biological activities. A probable mechanism of action underlying the recording of biological activities would considerably increase the scientific value of the manuscript. The discussion part needs to be more detailed. The entire section should be reconsidered, given these aspects.
Conclusion
The conclusion should accurately present the relevant findings, suggestions for improvement, and further research.
References
More adequate literature references are needed.
Author Response
Reviewer #2
General remarks:
The manuscript's authors aim to study the chemical diversity and biological activity of the essential oils and hydrolates obtained from aerial parts of different Crithmum maritimum ecotypes. The authors concluded that hydrolate from Croatian sea fennel is a good candidate for cosmetic applications as an anti-aging or preservative agent. The manuscript must be significantly reviewed.
Response: We would like to thank the Reviewer for the time, efforts and comments. We have considered all comments and suggestions and the manuscript has been modified accordingly.
Abstract
The abstract should be more precise and better organized. The aim of the study needs to be more accurate.
Response: Following the comment of the Reviewer, we improved the Abstract.
Introduction
The manuscript's introduction must be restructured with information relevant to the study's objectives. For example, the comparative analysis of publications studying essential oils vs. hydrolates, presented in the manuscript, needs to be sufficiently justified in the context in which the properties of hydrolases (in general) are comprehended. Hydrolates already have several applications in the cosmetic and pharmaceutical industries. Please revise.
Response: The introduction section has been modified according to the comment. Thank you.
Materials and method
How was the plant material used identified? Is there a voucher for each species studied? Where is it stored? Also, the authors should include information about the statistical analysis performed.
Response: Following the comment of the Reviewer, the M&M section has been improved/completed/modified.
Results and discussion
The scientific level could be better throughout the manuscript, especially in interpreting the results. Although the authors declare MIC determination (lines 162-164), the results are not in the results and discussions section. Please review! Also, the authors need to explain the causes of the recorded biological activities. A probable mechanism of action underlying the recording of biological activities would considerably increase the scientific value of the manuscript. The discussion part needs to be more detailed. The entire section should be reconsidered, given these aspects.
Response: We have deleted the two lines (162-164) because the MIC could not be reached due to the lack of matter (EO, hydrolate). The antioxidant activity of the samples was additionally discussed and related to the chemistry of the samples. The mechanism of action of the bacteriostatic effect of the plant EO or the hydrolates was not investigated and therefore not discussed.
Conclusion
The conclusion should accurately present the relevant findings, suggestions for improvement, and further research.
Response: Following the comment of the Reviewer, we improved the Conclusion section.
References
More adequate literature references are needed.
Response: The entire manuscript has been supplemented by several new/recent studies.
Reviewer 3 Report
Comments and Suggestions for Authors
To the esteemed authors:
I have carefully read the manuscript and have identified several deficiencies in the research that need to be addressed in order to improve the manuscript. Firstly, the Chemical Compound section should be separate from the Biological Activity Assay section. It is important to focus on the differences in chemical composition, as you have done in the antibacterial section. Additionally, some sections lack in-depth discussion and should be rewritten and improved. Please also correct the manuscript based on the comments provided below:
Line 105: Add a reference for the recommended storage temperature.
2.4.1: Use analysis of free radicals to measure antioxidant activity (only one ORAC assay is different). These methods confirm the antioxidant type I. There may be other antioxidant assays, such as metal chelating activity or reduction power, which show positive results and indicate antioxidant type II metabolites.
Line 156: Add the name of the paper from Disc Company.
Line 157: Why use 7.5 microliters? In the reference methods, 10, 15, and 20 microliters are used based on the paper size, ranging from 6 to a higher diameter.
Line 162-164: It is important to run the MBC measure, especially for the antibacterial test, specifically on E. coli.
Line 169-171: Double-check the amounts according to the reference again.
Line 208: Add the statistical analysis methods used.
Line 210-226: This section presents your hypothesis that the research was set up to confirm. This section should be transferred and combined into the introduction section.
Line 237: Also, include the sabinene amount in the two specimens.
Line 261-263: Sabinene is only high in the Croatian specimen, not in the French one.
Table 1: Explain if these data belong to one sample run or more. It is better to include standard deviation for the results and perform a t-test statistical analysis. Also, rearrange the RI.
3.1. Chemical profile of essential oils: You only report the other chemical components from literature. You didn't discuss why these amounts are different between the different site selected specimens. Are the environmental conditions or water chemical composition responsible?
Table 2: Same as Table 1.
Line 368: It is not acceptable to say that the results in the French sample of EO and hydrolysate compound are similar in dillapiole and γ-terpinene. Also, how can you say they "differ significantly"? You don't have any statistical analysis to confirm.
Line 375-380: You must conclude the differences between EO and hydrolate based on the water affinity of the compounds and the effects of temperature and extraction method. This section needs a deeper discussion.
Line 397: ...as are the?!
Line 398-402: This paragraph does not fit here after reporting negligible antioxidant results. It is similar to the introduction. The same goes for lines 403-411.
Line 403-411: You must first report the results of your research.
Antioxidant section: There is a problem with your research method. You didn't run a positive control to compare the results. Which data is fair or good? You can't conclude because you don't have a positive control.
Line 418: It is not true that ORAC in FRA EO and CRO EO showed any significance.
Line 421-423: You must discuss antioxidant differences based on the main chemical compounds you reported in the previous section.
Line 453: Scientific names must be in italic style. Correct this throughout the entire document.
Line 476: Add a conclusion based on the positive controls and possible future usage as cosmeceuticals.
Conclusion: Line 494: Mention which kind of antioxidants. Your results don't show any significant antiradical properties.
Comments on the Quality of English Language
English language need correction by a native editor system.
Author Response
Reviewer #3
To the esteemed authors:
I have carefully read the manuscript and have identified several deficiencies in the research that need to be addressed in order to improve the manuscript.
Firstly, the Chemical Compound section should be separate from the Biological Activity Assay section. It is important to focus on the differences in chemical composition, as you have done in the antibacterial section. Additionally, some sections lack in-depth discussion and should be rewritten and improved. Please also correct the manuscript based on the comments provided below:
Response: We would like to thank the Reviewer for the time and helpful comments that improved the quality and clarity of our manuscript. We have considered all comments and suggestions and the manuscript has been modified accordingly.
According to the suggestion the chemical compound section is now separated from the biological activity assay section. Also, the whole manuscript has been revised and esspecially the discussion.
Line 105: Add a reference for the recommended storage temperature.
Response: A reference has been added.
2.4.1: Use analysis of free radicals to measure antioxidant activity (only one ORAC assay is different). These methods confirm the antioxidant type I. There may be other antioxidant assays, such as metal chelating activity or reduction power, which show positive results and indicate antioxidant type II metabolites.
Response: By three methods (multiple method approach, DPPH, NO and ORAC) we investigated the activity of primary antioxidants (free radical terminators/scavengers, HAT-based), while the FRAP method was used to determine reducing power of the samples (ET-based) by the action of electron donating antioxidants. The discussion regarding the otianed results has been modified/improved.
Line 156: Add the name of the paper from Disc Company.
Response: Done
Line 157: Why use 7.5 microliters? In the reference methods, 10, 15, and 20 microliters are used based on the paper size, ranging from 6 to a higher diameter.
Response: We did not mention a reference method in the Materials & Methods section. Due to the low amount of EO and hydrolate, we could only use 7.5 microliters, not more.
Line 162-164: It is important to run the MBC measure, especially for the antibacterial test, specifically on E. coli.
Response: As mentioned above, the low amount of EO or hydrolate did not allow us to use a range of concentrations to measure MBC or MIC.
Line 169-171: Double-check the amounts according to the reference again.
Response: We have made a slight modification of Masuda et al. reference method for anti-tyrosinase bioassay, using tyrosine instead of L-DOPA, and 30 units tyrosinase/mL instead of 46 units/mL.
Line 208: Add the statistical analysis methods used.
Response: This has been added.
Line 210-226: This section presents your hypothesis that the research was set up to confirm. This section should be transferred and combined into the introduction section.
Response: The suggestion of the Reviewer has been accepted and this paragraph has been moved to the Introduction.
Line 237: Also, include the sabinene amount in the two specimens.
Response: The differences between the chemotypes of sea fennel can so far only be explained by the presence and content of dillapiol, which has been pointed out in this line (chemotype I with a content of 15–17% and chemotype II with 0–6%, Pateira et al., 1999). However, other compounds affecting the differences between chemotypes (including sabinene and limonene) are described above in lines 231-236.
Line 261-263: Sabinene is only high in the Croatian specimen, not in the French one.
Response: Agree. This has been corrected (deleted).
Table 1: Explain if these data belong to one sample run or more. It is better to include standard deviation for the results and perform a t-test statistical analysis. Also, rearrange the RI.
Response: The GC-MS analysis of the samples was performed in duplicate and statistical deviations are now provided in both Tables. Thank you for the comment.
3.1. Chemical profile of essential oils: You only report the other chemical components from literature. You didn't discuss why these amounts are different between the different site selected specimens. Are the environmental conditions or water chemical composition responsible?
Response: This part of discussion has been improved. Thank you for the comment.
Table 2: Same as Table 1.
Response: The GC-MS analysis of the samples was performed in duplicate and statistical deviations are now provided in both Tables. Thank you for the comment.
Line 368: It is not acceptable to say that the results in the French sample of EO and hydrolysate compound are similar in dillapiole and γ-terpinene. Also, how can you say they "differ significantly"? You don't have any statistical analysis to confirm.
Response: Agree. This part has been modified.
Line 375-380: You must conclude the differences between EO and hydrolate based on the water affinity of the compounds and the effects of temperature and extraction method. This section needs a deeper discussion.
Response: We thank the reviewer for this comment and according to his suggestion we have improved this part of discussion.
Line 397: ...as are the?!
Response: Corrected. Thank you for noticing this.
Line 398-402: This paragraph does not fit here after reporting negligible antioxidant results. It is similar to the introduction. The same goes for lines 403-411.
Response: According to the Reviewer comment, this paragraph(s) has/have been deleted.
Line 403-411: You must first report the results of your research.
Response: Corrected.
Antioxidant section: There is a problem with your research method. You didn't run a positive control to compare the results. Which data is fair or good? You can't conclude because you don't have a positive control.
Response: We have added the results for positive control (gallic acid).
Line 418: It is not true that ORAC in FRA EO and CRO EO showed any significance.
Response: Agree. As there is no statistical difference we have re-formulated this sentence.
Line 421-423: You must discuss antioxidant differences based on the main chemical compounds you reported in the previous section.
Response: The discussion has been improved/corrected.
Line 453: Scientific names must be in italic style. Correct this throughout the entire document.
Response: Corrected and checked.
Line 476: Add a conclusion based on the positive controls and possible future usage as cosmeceuticals.
Response: We have discussed the anti-ageing activities relatively to those of the positive controls, and considering the huge dilution of pure EOs that was necessary to obtian the IC50 in these bioassays.
Conclusion: Line 494: Mention which kind of antioxidants. Your results don't show any significant antiradical properties.
Response: Following the comment of the Reviewer, we improved the Conclusion section. Thank you.
Round 2
Reviewer 2 Report
Comments and Suggestions for Authors
Good luck in your future research projects.
Author Response
We thank the reviewer for his careful reading of the manuscript, constructive and positive remarks.
Reviewer 3 Report
Comments and Suggestions for Authors
Dear Authors
The revised manuscript have good corrections. I believe it is better you mentioned about little amounts of the specimen in the antibacterial section of the method or results that cause you cant do complete analysis (7.5 microliter load on the paper disc and MBC test).
Comments on the Quality of English Language
Minor corrections is need for enhance the manuscript quality.
Author Response
We thank the reviewer for his careful reading of the manuscript, constructive and positive remarks.
We have added the requested comment on the sample volume used in antimicrobial testing and reviewed the entire manuscript (English proofreading with Instatext).